# Coffee Pulp: A Sustainable and Affordable Source for Developing Functional Foods

Angélica Pérez Calvo * , Nelson Paz Ruiz and Zuly Delgado Espinoza

Facultad de Ingeniería, Corporacion Universitaria Comfacauca-Unicomfacauca, Cl 4 N. 8-30,
Popayán 190001, Cauca, Colombia
* Correspondence: angelicaperez@unicomfacauca.edu.co

**Abstract:** Coffee cultivation in the Department of Cauca, Colombia, is vital for much of the department's economy. Coffee processing generates waste, such as pulp, which accounts to about 40% of the fresh fruit. Currently, in most cases in Cauca, coffee byproducts are discarded, and, in other cases, the coffee pulp is used for fertilizers, generating environmental problems due to its decomposition. This research aims to design a process for supplying a functional food in the form of an energy bar based on coffee pulp and other components of the region, such as quinoa and panela, helping to mitigate the environmental impact of coffee production. Our research included four phases. First, we determined the study area; in the second phase, we studied an energy bar's nutritional and physical characteristics. Then, the requirements and specifications of the bar were defined, and the authors documented the process diagram, variables within the process, and the quality plan. Finally, the authors conducted experiments to determine optimal mixing proportions. From the experiment with the mixtures, we found a formulation that satisfies the needs and specifications of the bar which is composed of 50% cereals, 30% panela syrup, and 20% coffee pulp. The selected formulation's qualitative properties (organoleptic, chemical, and microbial) are acceptable for human consumption and provide high energy content of 365.21 and 291.56 kcal/100 g for the energy bar and coffee pulp raisins, respectively.

**Keywords:** coffee; experimental design; circular economy; environmental impact; sustainability; value added

## 1. Introduction

Colombia is the third largest coffee producer worldwide; activities surrounding coffee production and processing generate more jobs in the country than any other economic activity. According to a 2019 report from the National Federation of Coffee, the industry supports 785,000 jobs. In other words, growing coffee is an integral national economic development source [1].

The Department of Cauca, Colombia, includes 42 potential municipalities, most of which depend on coffee production, being the fourth largest producer in Colombia. In 33 Cauca municipalities, 91,000 families cultivate coffee, with a total of 92,000 hectares (ha) planted, producing 1,500,000 sacks of dry parchment coffee. A total of 33% of the rural population derives its income from growing coffee, generating approximately 65,000 direct jobs. It is worth noting that 99.40% of the families are small coffee growers, 0.10% are medium coffee growers, and 0.50% are large coffee growers [2].

In coffee processing, coffee residues constitute a major source of contamination and environmental problems. For this reason, researchers search for ways to use the coffee residues to produce food supplements, beverages, vinegar, biogas, pectin, pectic enzymes, proteins, fertilizers, and animal feed, among others [3].

Rodriguez and collaborators in 2010 showed that every kilogram (kg) of dry coffee generated approximately one kg of pulp waste that is currently discarded and causes

contamination due to the decomposition of organic solids, producing gas emissions and leachates. Please note that 162,900 tons of poorly managed pulp can cause pollution equivalent to that generated by a population of 868,736 inhabitants in excreta and urine during a year [4,5]. The same study found that coffee residues have interesting nutritional properties. Rodriguez and Zambrano reported that on a dry basis, it contains 17.31% reducing sugars and 7% protein [6].

At the same time, people are seeking quality foods with high caloric and nutritional content. So, the market has incorporated foods that are easy to access and consume (energy bars and functional, fortified, and enriched foods) and can be used in emergency conditions to provide health benefits [5].

Several authors have studied coffee by-products' functional properties and structural characteristics to verify if they can be reused in functional foods. Ballesteros reported that coffee pulp is a lignocellulosic element with 51.50% carbohydrates polymerized to its cellulose fractions and 40.45% hemicellulose. Moreover, the pulp has excellent water and oil retention capacity and high anti-oxidant potential; for this reason, it has the potential to be used as a raw material for food products with high nutritional value [7]. The novelty of this study is to take advantage of this potential uses of coffee residue to generate an energy food, thereby creating new opportunities for agricultural and rural development.

## 2. Materials and Methods

The study includes five phases: (1) definition and diagnosis of the study area; (2) characterization of the production process; (3) design of a quality plan; (4) application of an experimental mixture design; and (5) proximate and microbiological analysis.

### 2.1. Study Zone

The study was carried out in the Department of Cauca, specifically in the municipalities that grow coffee, sugarcane, and quinoa; they were identified through institutional web pages, such as the National Federation of Coffee Growers of Cauca, and the report of Municipal Agricultural Evaluations (EVA), issued by the Ministry of Agriculture.

### 2.2. Product Characterization

The study characterized the product in terms of the nutritional properties of the raw materials used in the production of energy bars, using institutional web pages, such as the Food and Agriculture Organization of the United Nations (FAO) [8].

### 2.3. Quality Plan

Quality planning guarantees the desired characteristics of the product, for which it is essential to define the project, define product specifications and needs, determine the process, refine specifications of the product, establish controls at each stage of production, and transfer the information needed to all stakeholders [9].

Juran Matrix: To define the requirements and specifications of the proposed energy bar, a Juran matrix was developed, which evaluates the relationship between the expected purposes of the product and the customer's needs [9]. These factors were ordered according to their importance from highest to lowest; likewise, a weighting was given to the relationship between the factors, from 10 to 30 points, 10 being less significant and 30 more significant [10].

Production Process: As reported by Betancourt and Verduga et al. [11], the production process was designed. Then, tests were carried out in the kitchen laboratory, where each stage of the process was analyzed to determine quality control in each one of them [12].

Control Metrics: Once the control variables were defined and metrics were established to verify compliance, control charts were formatted. There are two types of charts. The first type considers variation for continuous variables (weight, volume, length, etc.) in terms of averages (X_bar). The second type considers the mean ranges (R), standard deviations

(S), and individual measures (X). Both types are used to assess product quality and, thus, minimize defective units (np), the number of defects (c), and some defects per unit (u) [13].

Spreadsheet or Analysis: The information collected was documented in an analysis spreadsheet that included the subject of control, units of measurement, types of metrics, expected objective of the variables, frequency of measurement, sample size, corrections in case of non-compliance and those responsible for the execution and approval of the improvements [9,14].

### 2.4. Experimental Design

An experimental design of the mixtures was implemented to determine the exact proportions the ingredients (cereals, panela syrup, and coffee pulp) should have to guarantee a quality and consistent final product. The formula for the model is as follows,

$$Z_m a x = Y_1, Y_2, Y_3 \tag{1}$$

where $Y_1$ is the weight, $Y_2$ is the Kcalories, and $Y_3$ is the sense acceptance. In addition, the following two factors influence the response variables:

$$X_1 = \% \text{ anela syrup}$$
$$X_3 = \% \text{ Coffee Pulp.}$$

Constraints on influencing factors include the following:

$$50\% \leq X_1 \leq 60\%$$
$$20\% \leq X_2 \leq 30\%$$
$$10\% \leq X_1 \leq 20\%$$

The chef Juan Carlos Ramos Velasco, a teacher at the Corporación Universitaria Comfacuaca–Unicomfacuaca defined restrictions for the product. He indicated that cereal content should be between 50% and 60%. If the cereal content is lower or higher than those limits, quality would be affected, creating either a flaccid or overly hard consistency, depending on the case. The panela syrup content should be between 20% and 30% because, if it exceeds 30%, the product would be too sweet, and if it is lower than 20%, the bar would not be consistent. Finally, the coffee pulp should be between 10% and 20% because if it exceeds these values its flavor could overpower the other ingredients, while content lower than 10% would be insignificant in the bar. Once the mathematical model was defined, it ran in Minitab software. The model was entered into the software, indicating that it is an extreme vertices design. The program yielded seven formulations according to the information executed as shown in Table 1.

**Table 1.** Model formulation.

| Formulation | Cereals (%) | Panela Syrup (%) | Coffee Pulp (%) |
|:---:|:---:|:---:|:---:|
| 1 | 0.53333 | 0.28333 | 0.18333 |
| 2 | 0.50000 | 0.30000 | 0.20000 |
| 3 | 0.58333 | 0.23333 | 0.18333 |
| 4 | 0.60000 | 0.20000 | 0.20000 |
| 5 | 0.56667 | 0.26667 | 0.16667 |
| 6 | 0.60000 | 0.30000 | 0.10000 |
| 7 | 0.58333 | 0.28333 | 0.13333 |

Next, the seven formulations were prepared, and the response variables were analyzed in each to evaluate the influence of the components on the following variables.

Yield:For this yield analysis, the bars were left to stand for 4 h until they had a resistant consistency; then, each of the formulations was weighed, using a calibrated precision scale.

Caloric Contribution: The nutritional content of each of the raw materials was considered, noting that 1 gram (g) of fat contributes 9 Kcalories, and 1 g of protein and carbohydrates contribute 4 Kcalories each [8].

Sensory Acceptance: A sensory analysis was performed using a 9-point hedonic test, where 1 was extremely dislike, 2 was strongly dislike, 3 was dislike, 4 was slightly dislike, 5 was neither like nor dislike, 6 was slightly like, 7 was like, 8 was like very much, and 9 was like extremely. The test was designed to obtain results in terms of overall product acceptance [15]. Likewise, a JART test was used to analyze the characteristics of cohesiveness, hardness, sweetness, and chewiness. We worked with a panel of 8 experts, each presented with 7 samples coded with 3 digits [16].

Simulation of Experimental Model: With the results obtained, the simulation was carried using a random component response surface model where the the yield, caloric contribution, and sensory acceptance of each formulation was evaluated [16], and a p-value (PV) of less than 0.05 was used to determine if there was a significant influence on the variables of interest [17,18], Based on the simulation, the following hypotheses were established.

$$H1: X_{(1,2,3)} \; Significantly \; influences \; Y_{(1,2,3)}$$
$$H0: X_{(1,2,3)} \; Does \; not \; Significantly \; influences \; Y_{(1,2,3)}$$

### 2.5. Proximal and Microbiological Analysis

The samples were prepared in the foods laboratory of the Corporación Universitaria Comfacauca–Unicomfacauca, according to the flowchart for producing energy bars by adding coffee pulp raisins, as shown in Figure 1.

| | Requirements. / Purposes | 5 High energy content | | 2 100% natural | | 1 Labeled and labeled packaging | | 6 Nutritional specifications and textures | | 7 Organoleptic characteristics | | 3 Bar presentation | | 4 economic | | Total |
|---|---|---|---|---|---|---|---|---|---|---|---|---|---|---|---|---|
| 6 | Quality | 30 | 900 | 30 | 360 | 30 | 180 | 30 | 1.080 | 30 | 1.260 | 30 | 540 | 10 | 240 | 4,560 |
| 3 | Innovation | 20 | 300 | 30 | 180 | 10 | 30 | 10 | 180 | 10 | 210 | 10 | 90 | 10 | 120 | 1,110 |
| 2 | Competitiveness | 20 | 200 | 30 | 120 | 20 | 40 | 30 | 360 | 30 | 420 | 10 | 60 | 30 | 240 | 1,440 |
| 1 | cost reduction | 10 | 50 | 10 | 20 | 10 | 10 | 20 | 120 | 10 | 70 | 20 | 60 | 30 | 120 | 450 |
| 3 | Performance and productivity | 10 | 150 | 10 | 60 | 10 | 30 | 20 | 360 | 20 | 420 | 10 | 90 | 20 | 240 | 1,350 |
| 5 | Profitability | 20 | 500 | 20 | 200 | 20 | 100 | 30 | 900 | 30 | 1050 | 20 | 300 | 20 | 400 | 3,450 |
| | Total | 2,100 | | 940 | | 390 | | 3,000 | | 3,430 | | 1,140 | | 1,360 | | 12,360 |

**Figure 1.** Energy bars prioritization matrix.

A total of 200 grams of each product was prepared for microbiological analysis and 400 grams for proximal analysis. The energy bar was vacuum-packed in bags, and the coffee pulp raisins were vacuum-packed in glass jars to guarantee their shelf life until the tests were carried out.

The energy bar and the coffee pulp raisins were sent to a certified laboratory in the country (SymlaB Bioindustrial laboratory) where the parameters shown in Table 2 were analyzed.

**Table 2.** Analysis methods.

| LABORATORY TEST | ANALYSIS: | METHOD |
|---|---|---|
| MICROBIOLOGICAL ANALYSIS | Total mesophilic aerobic count CFU/g-mL | AOAC 966.23ED. 21:2019 |
|  | Molds and yeasts count CFU/g-mL | ISO 21527-2: 2008 |
|  | Total content MPN /g-mL | ICMSF:2000 Method 1, Volume 1, Ed. 2:2000 |
|  | Fecal Content MPN 45 °C/g-mL | ICMSF:2000 Method 1, Volume 1 Ed. 2:2000 |
|  | Coagulase-positive Staphylococcus count CFU/g-mL | UNE EN ISO 6888-1 2000 |
|  | Detection of Salmonella in 25 g | Norma ISO 6579-1:2017 |
|  | Mold and yeast counts UFC/g-mL | SO 21527-2: 2008 |
| PROXIMAL ANALYSIS | Humedad y materia volátil | PRO-AYS-057 V0 2021-07- 19 Determination of total solids and drying losses at 103 °C y 130 °C |
|  | ASH | PRO-AYS-067 V0 2021-07- 19 Ash Determination 550 °C |
|  | Total grasses | PRO-AYS-067 V0 2021-07- 19 Ash Determination 550 °C |
|  | Protein total (%N x 6.25) | PRO-AYS-055 V0 2021-07- 19 Protein determination according to ISO 1871 |
|  | Total carbohydrates | Calculation |
|  | Total calories |  |

## 3. Results and Discussion

### 3.1. Location of the Study Population and Characteristics of Inputs

According to the pattern of coffee growing in Cauca (Table 3), coffee production in the Department of Cauca for the year 2020 was approximately 130,625,000 kg; considering that the pulp is equivalent to 40% of the fresh beans, the amount of pulp is 52,250,000 kg. The main coffee growers are the municipality of Tambo, with a total of 8155 ha planted and production of dry parchment coffee of 10,250,000 kg, and the municipality of Piendamó with 7.721 ha planted, from which 1,250,000 kg of coffee is obtained.

**Table 3.** Municipalities with the highest participation in coffee growing in the Department of Cauca.

| Municipality | Area Cultivated Hectares | Tree Production | Production in Kilograms | Jobs Generated | Harvest Value in Millions of Pesos |
|---|---|---|---|---|---|
| Cajibío | 6.873 | 660,000 | 8,250,000 | 4.811 | 66,000 |
| Caldono | 4.789 | 640,000 | 8,000,000 | 3.353 | 64,000 |
| El Tambo | 8.155 | 820,000 | 10,250,000 | 5.708 | 82,000 |
| Inzá | 5.095 | 570,000 | 7,125,000 | 3.566 | 57,000 |
| Morales | 6.827 | 840,000 | 10,500,000 | 4.779 | 84,000 |
| Páez | 5.242 | 630,000 | 7,875,000 | 3.669 | 63,000 |
| Piendamó | 7.721 | 1,000,000 | 12,500.000 | 5.405 | 100,000 |
| Timbío | 4.095 | 540,000 | 6,750,000 | 2.866 | 54,000 |

Coffee pulp possesses various medicinal and anti-oxidant properties thanks to its nutritional content (see Table 4). Among its energy sources are proteins, with a yield of ap-

proximately 10.63 g per 100 g of product on a dry basis, fats with 5.78 g, and carbohydrates providing about 45.67 g. It is also classified as an energetic and anti-oxidant product thanks to its caffeine percentage, which is 2.26%.

**Table 4.** Nutritional composition of coffee pulp.

| Components | Contents |
| --- | --- |
| Proteins (g) | 10.63 |
| Fats (g) | 5.78 |
| Carbohydrates (g) | 45.67 |
| Ash (g) | 9.58 |
| Crude fiber (g) | 36.07 |
| Caffeine % | 2.26 |
| Humidity % | 12.05 |

Panela is a natural sugar substitute which comes from sugarcane juice, its only ingredient. It is mainly used as a sweetener, but it is also used in compacted molds used in baking [19]. In addition to sucrose, it contains significant glucose, fructose, proteins, minerals, and vitamins. In Cauca, 38 municipalities grow sugarcane, representing 90.50% of the Department's total [20].

Panela is a food rich in nutrients, minerals, and vitamins; a 100 g serving of panela contains 0.39 to 1.13 g of protein, 0.13 or 0.15 g of fat, and 75 to 84.48 g of carbohydrates; these nutritional contributions generate an energy source of approximately 351 Kcal [19].

Regarding cereals, the Department of Cauca grows quinoa, a cereal rich in fiber and minerals, with production in six municipalities. The Municipal Agricultural and Livestock Evaluations (EVA) report for 2020 noted that the Department produced 249,450 kg of quinoa [20].

Quinoa is a gluten-free pseudocereal, which is an excellent source of carbohydrates, protein, fiber, vitamins, and minerals; it is one of the few foods with a nutritional content close to the human nutrition standards established by FAO [21]. Every 100 g of the product has approximately 60 g to 71 g of carbohydrates and 13 g to 20 g of protein. It also contains 6.10 g of unsaturated fats, and the energy obtained is around 350 Kcal [22].

### 3.2. Production Process Characteristics

Functional foods do not have any consumption restrictions. However, certain populations with special nutritional needs do have consumption restrictions [23].

Nutritional and Physical Characteristics: A study developed by Córdova in 2018 reported that cereal bars must have between 10% and 12% of humidity, and their sugar percentage cannot be higher than 55%. Most bars on the market contain 60–80% carbohydrates, 3–24% fat, 4–15% protein, and 370–490 calories per 100 g; such products are considered to have high energy potential. The literature indicates that the most acceptable bars on the market are approximately 40 g; Table 5 presents the approximate nutritional values of an energy bar this size [24].

**Table 5.** Proximal composition of a 40-gram commercial energy bar.

| Components | Contents |
| --- | --- |
| Energy value (kcal) | 110–462 |
| Carbohydrates (g) | 11.80 |
| Proteins (g) | 10.30 |
| Greases (g) | 3 |
| fiber (g) | 4.50 |
| water (g) | 9.80 |
| Ashes (g) | 0.04 |

Design of a product quality plan: The results from the Juran matrix (Figure 1) determined that the most significant purposes are quality, with a score of 4.560, and profitability, with 3.450 points. As for the specifications, compliance with nutritional specifications was most important, with 3.000 points, and compliance with organoleptic characteristics, with 3.430 points.

Based on this, the product process diagram was defined (Figure 2), which starts at the raw material reception stage and ends at the finished product storage stage.

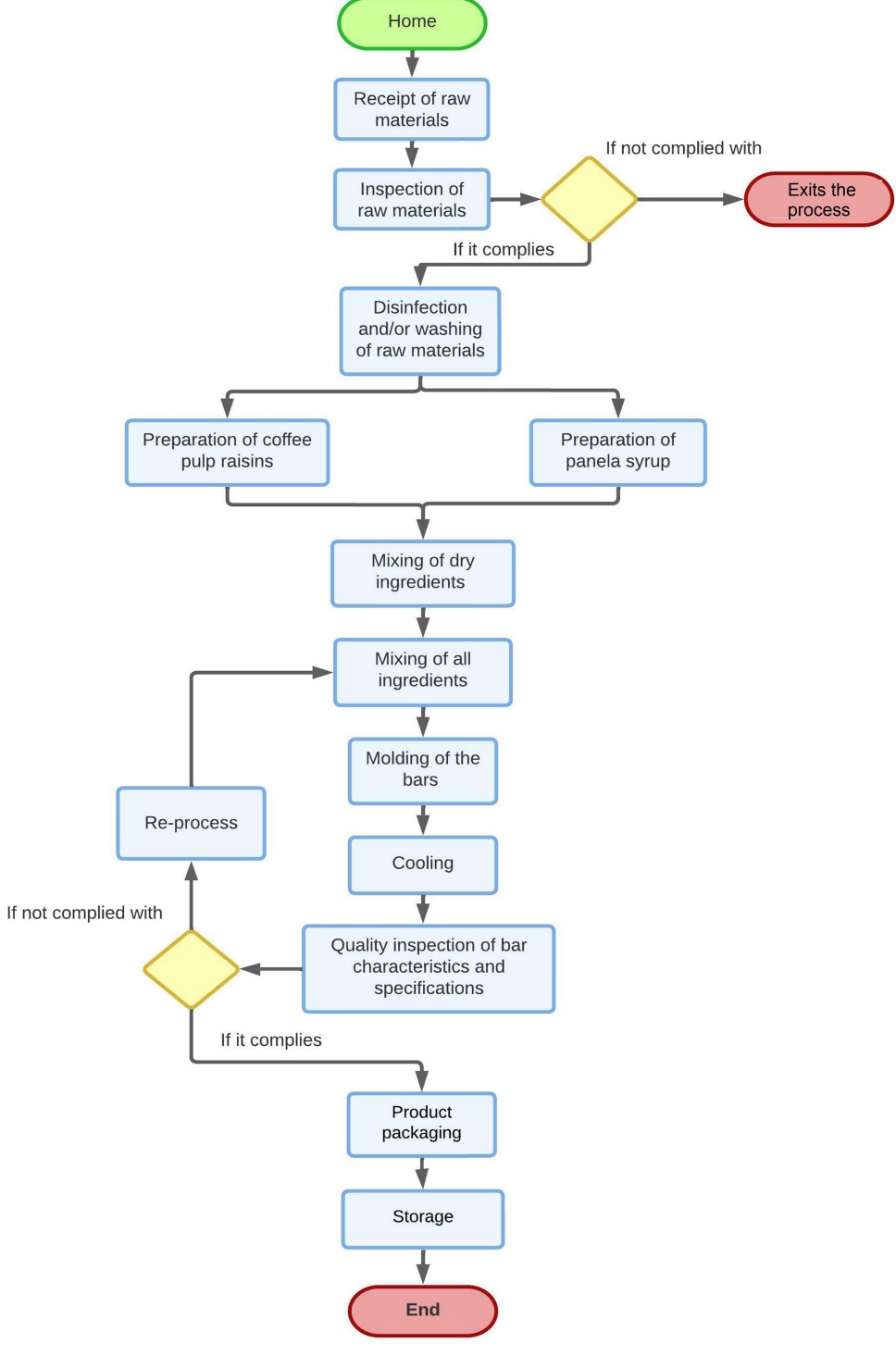

**Figure 2.** Flowchart for the production of energy bars with coffee pulp addition.

Quality plan documentation: In the analysis sheet of the quality plan,a quality inspection is proposed at the reception of raw materials using simple acceptance sampling; the raw materials will be controlled using IRM control charts and instruments, such as a stopwatch. Once the final product is obtained, the organoleptic characteristics will be inspected using a U control chart, the weight of the packaging will be monitored with an X_bar control chart, and the bars that pass the test will be taken to storage, where finally, the temperature will be controlled by means of an IRM chart.

### 3.3. Statistical Analysis of Experimental Design

The experimental design that was implemented was of mixtures by extreme vertices; once the mathematical model was defined, it was executed in the Minitab software described in the methodology. As a result, the experimental region shown in Figure 3 was obtained, where the gray triangle represents the region restricted by the limitations, within which the treatments provided by the model are located. The red points represent the vertices, which correspond to formulations 2, 4, and 6, and the green point indicates the central point, i.e., the mixture where the proportions of the components are the averages of the proportions of the corresponding vertices, which, in this case, is formulation 5. Finally, the blue points indicate the axial points, located exactly halfway between the central point and a vertex, reflecting formulations 1, 3, and 7.

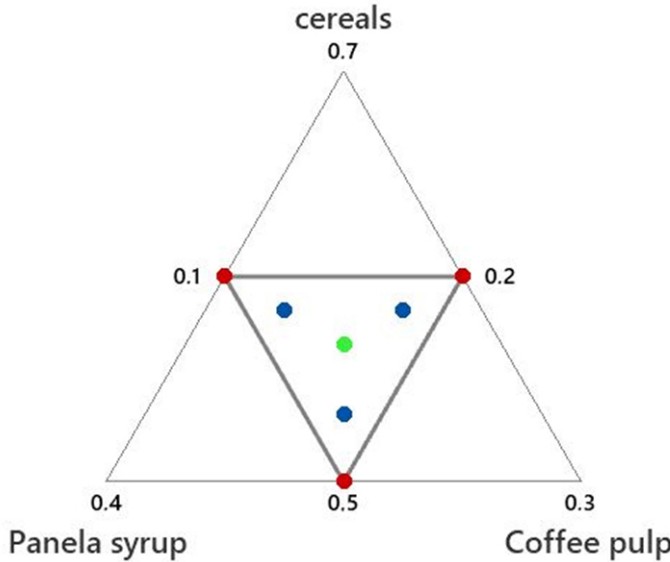

**Figure 3.** Experimental study region.

Table 6 shows the results obtained for the variables of interest by formulation. The best performance was observed in formulations 2, 4, and 6, which were greater than 99%. On the other hand, formulation 6 has the highest caloric content with 240.53 Kcalories. Regarding sensory acceptance, formulation 2 scored the best (7.69 points).

Yield: The results analyzed using the mixture design established that the quadratic model best fits the yield data with a determination coefficient of 99.46 (Table 7) and a *p*-value of 0.0542 (Table 8). It was also analyzed that when there is high participation of cereals and coffee pulp, a higher bar yield is achieved, but with higher values of panela syrup, the yield obtained is very low (Figure 4). In other words, as the percentage of panela syrup increases, the yield decreases, which indicates that this ingredient influences the low yield of the bar because the syrup loses moisture during the cooling process, which affects the final weight of the bar.

**Table 6.** Experimental results of yield, caloric intake, and sensory acceptance of each formulation.

| Formulation | Cereals (%) | Syrup of Panela (%) | Coffee Pulp (%) | Yield (%) | Quantity Kcalories | Sensory Analysis (Points) |
|---|---|---|---|---|---|---|
| 1 | 0.533 | 0.283 | 0.183 | 99.000 | 234.940 | 7.219 |
| 2 | 0.500 | 0.300 | 0.200 | 99.500 | 232.580 | 7.688 |
| 3 | 0.583 | 0.233 | 0.183 | 97.600 | 238.040 | 4.750 |
| 4 | 0.600 | 0.200 | 0.200 | 88.300 | 238.780 | 7.656 |
| 5 | 0.567 | 0.267 | 0.167 | 98.500 | 237.300 | 6.344 |
| 6 | 0.600 | 0.300 | 0.100 | 99.400 | 240.530 | 6.719 |
| 7 | 0.583 | 0.283 | 0.133 | 98.900 | 238.910 | 6.688 |

**Table 7.** Model adjustment for yield.

| Model | ES | R-Squared | R-Square— Adjusted Square |
|---|---|---|---|
| Lineal | 2.37925 | 76.71 | 65.07 |
| Quadratic | 0.72243 | 99.46 | 96.78 |
| Special cubic | | 100 | 0 |

**Table 8.** ANOVA analysis for yield.

| Source | Sum of Squares | GL | Mean Square | Reason F | *p*-Value |
|---|---|---|---|---|---|
| Average | 66.2905 | 1 | 66.2905 | | |
| Lineal | 74.5852 | 2 | 37.2926 | 6.59 | 0.0542 |
| Quadratic | 22.1215 | 3 | 7.37383 | 14.13 | 0.1899 |
| Special cubic | 0.5219 | 1 | 0.5219 | | |
| Error | $4.0099 \times 10^{-12}$ | 0 | | | |
| Total | 66.3877 | 7 | | | |

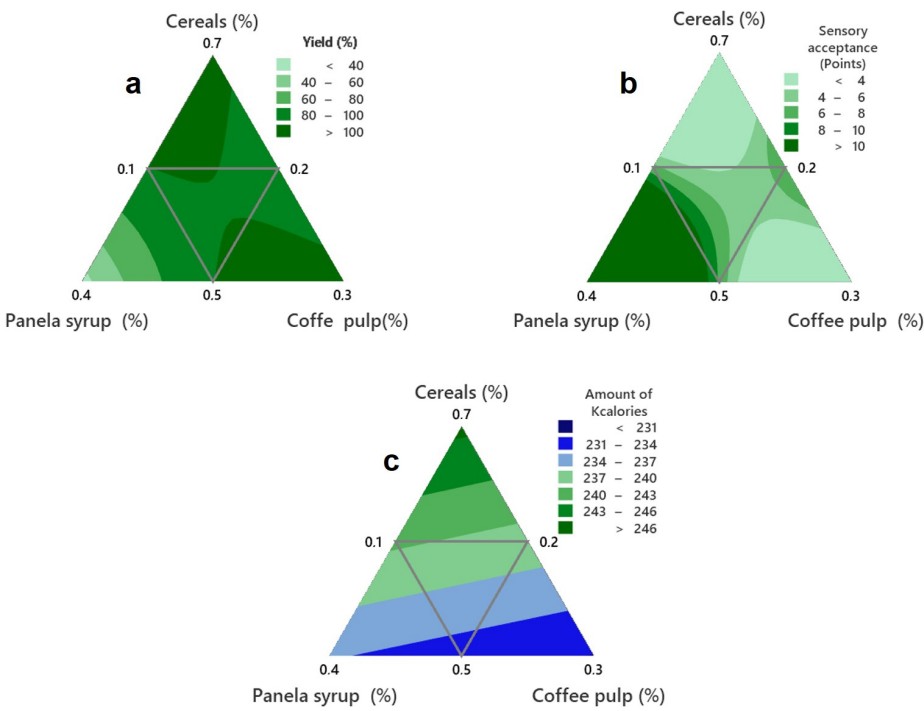

**Figure 4.** Contour plot of the influence of the components on yield (**a**), sensory acceptance (**b**), and caloric intake (**c**).

Caloric value: Figure 4 shows that when there are high percentages of panela syrup and especially of coffee pulp, the caloric value is very low; however, when the percentages of cereals in the formulation are quite high, the amount of energy provided is high. Cereals and caloric value have a directly proportional relationship since when the participation of cereals increases, the caloric value increases in the same proportion. On the other hand, panela syrup and coffee pulp have an inversely proportional relationship with the caloric value since, when these ingredients increase their participation in the formulation, the caloric value decreases (Figure 5). The linear model fits adequately for this variable of interest since the variance analysis showed the model's significance with a *p* value > 0.05. (see Tables 9 and 10), that is, the null hypothesis is rejected.

**Table 9.** Model adjustment for calorific value.

| Model | ES | R-Squared | R-Square— Adjusted Square |
|---|---|---|---|
| Lineal | 0.0024119 | 100 | 100 |
| Quadratic | 0.0026969 | 100 | 100 |
| Special cubic | | 100 | 0 |

**Table 10.** ANOVA analysis for calorific value.

| Source | Sum of Squares | GL | Mean Square | Reason F | *p*-Value |
|---|---|---|---|---|---|
| Average | 394.170 | 1 | 394.170 | | |
| Lineal | 43.6109 | 2 | 21.8055 | 3.7484 | 0 |
| Quadratic | $1.600 \times 10^{-5}$ | 3 | $5.33191 \times 10^{-6}$ | 0.73 | 0.6728 |
| Special cubic | $7.27328 \times 10^{-6}$ | 1 | $7.27328 \times 10^{-6}$ | | |
| Error | $-3.69552 \times 10^{-11}$ | 0 | 0 | | |
| Total | 394.213 | 7 | | | |

Sensory analysis: The sensory score is significant at the highest concentration percentages of panela syrup, while at the highest percentages of cereals and coffee pulp, the sensory score is minimal. When the panela syrup increases its percentage of participation, the sensory acceptance score also increases, while when the participation of cereals and coffee pulp increases, the sensory score decreases, as shown in Figure 5 and Tables 11 and 12 ANOVA. In conclusion, the panela syrup significantly influences the overall acceptance of the bar, while the other ingredients have no significant influence. The model that best fits the sensory analysis data is the quadratic model since the coefficient of determination is the most significant compared to the other models, which is 97.44% (see Table 11).

**Table 11.** Model adjustment for sensory acceptance.

| Model | ES | R-Squared | R-Square— Adjusted Square |
|---|---|---|---|
| Lineal | 1.14576 | 13.7 | 0 |
| Quadratic | 0.3943 | 97.44 | 84.67 |
| Special cubic | | 100 | 0 |

**Table 12.** ANOVA analysis for sensory acceptance.

| Source | Sum of Squares | GL | Mean Square | Reason F | *p*-Value |
|---|---|---|---|---|---|
| Average | 316.411 | 1 | 316.411 | | |
| Lineal | 0.833739 | 2 | 0.41687 | 0.32 | 0.7447 |
| Quadratic | 5.09561 | 3 | 1.69854 | 10.93 | 0.215 |
| Special cubic | 0.155572 | 1 | 0.155472 | | |
| Error | $4.9063 \times 10^{-13}$ | 0 | 0 | | |
| Total | 322.496 | 7 | | | |

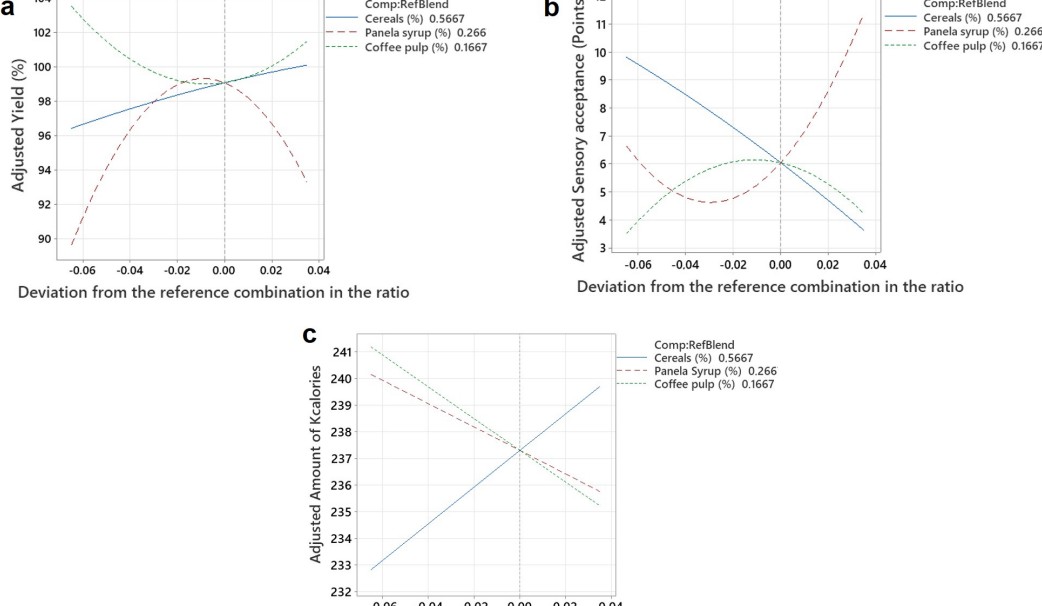

**Figure 5.** Cox response tracking plot for yield (**a**), sensory acceptance (**b**), and caloric intake (**c**).

Optimal region: Figure 6 shows the optimal region according to the objective function of each of the response variables, where the optimal region is oriented to the highest values of the panela syrup, which is close to a sensory acceptance of 9 points and a yield of 100%; however, in terms of kilocalories, we see the green dotted line (indicating the maximum expected energy of the bar), which is close to the highest limit of the cereals.

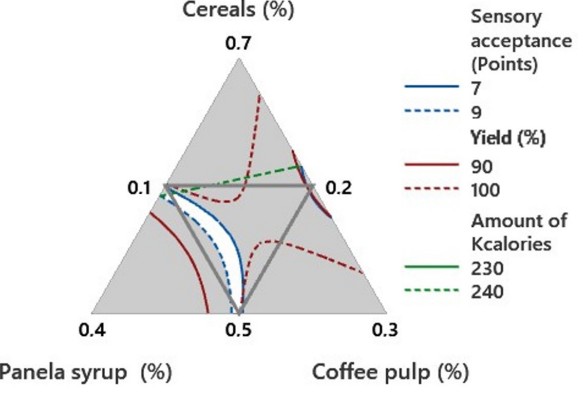

**Figure 6.** Contour plot intercepting sensory analysis, yield, and calorific value.

After analyzing the results and running the model in the Minitab software, it was found that the optimal formulation for the preparation of the bars is formulation number 2, composed of 50% bowls of cereal, 30% panela syrup, and 20% coffee pulp (Figure 7), which guarantees compliance with the specifications.

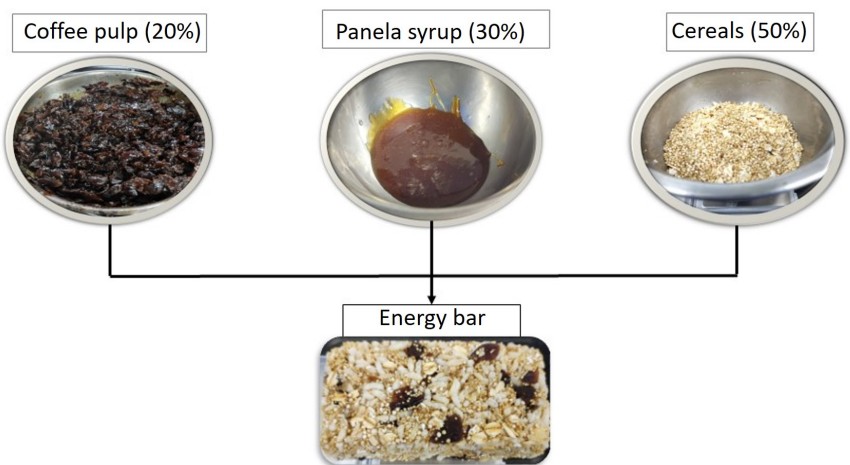

**Figure 7.** Energy bar with added coffee pulp. Source: the authors.

The next step was the experimental development of the energy bars in the kitchen laboratories of the Comfacauca Unicomfacauca University Corporation, to which proximal and microbiological analyses were carried out, as was done with the coffee raisin.

### 3.4. Microbiological and Nutritional Analysis

The samples did not show growth of yeast, fungi, mesophiles, staphylococci, total coliforms, fecal coliforms, and Salmonella in the microbiological analysis according to Resolution 1407 of 2022 of the Ministry of Health, which establishes the microbiological criteria that food and beverages intended for human consumption must meet, both products are below the permitted limits, considering as safe and suitable for consumption (see Table 13).

**Table 13.** Microbiological analysis of energy bars and coffee pulp raisins.

| SAMPLE | ANALYSIS | METHOD | SPECIFICATION | RESULT |
|---|---|---|---|---|
| EN-ERGY BAR | Total mesophilic aerobic count CFU/g-mL | AOAC 966.23 ED. 21:2019 | 5.000 (m)–10.000 (M) | 30 (±1 UFC ) |
| | Molds and yeasts count CFU/g-mL | ISO 21527-2: 2008 | 1.000 (m)–2.000 (M) | <100 (±1 UFC ) |
| | Total coliforms MPN /g-mL | ICMSF:2000 Method 1, Volume 1, Ed. 2:2000 | 9 (m)–110 (M) | 3 |
| | Fecal Coliforms MPN 45 °C/g-mL | ICMSF:2000 Method 1, Volume 1 Ed. 2:2000 | 3 | 3 |
| | Coagulase-positive Staphylococcus count CFU/g-mL | UNE EN ISO 6888-1 2000 | 100 | 100 (±1 UFC ) |
| | Detection of Salmonella in 25 g | ISO 6579-1: 2017 | AUSENCIA | AUSENCIA |
| COFFEE PULP RAISINS | Mold and yeast counts UFC/g-mL | SO 21527-2: 2008 | 10 (m)–100 (M) | 10 (±1 UFC ) |

In the proximal analysis (Table 14), the samples present a high content of total carbohydrates, a low contribution of protein and total fat, as well as a significant source of energy, being 365.21 and 291.56 kcal/100 g for the energy bar and coffee pulp raisins, respectively, compared to the results of the analysis with the average values of a commercial energy bar (Table 5) [25]. The study affirmed that the energy bar with coffee pulp addition is a product with high nutritional and energetic value that can satisfy the need for caloric intake.

**Table 14.** Proximal analysis of energy bars and coffee pulp raisins.

| SAMPLE | ANALYSIS | UNIT | SPECIFICA-TION | RE-SULT | METHOD |
|---|---|---|---|---|---|
| EN-ERGY BAR | MOISTURE AND VOLATILE MATTER | g/100 g | N/A | 8.57 | PRO-AYS-057 V0 2021-07- 19  Determination of total solids and drying losses at 103 °C y 130 °C |
| | ASH | g/100 g | N/A | 0.89 | PRO-AYS-067 V0 2021-07- 19  Ash Determination 550 °C |
| | TOTAL GRASES | g/100 g | N/A | 0.61 | PRO-AYS-113 V0 2021-07- 19  Determination of fat by ethereal extract |
| | TOTAL PROTEIN (%N x 6.25) | g/100 g | N/A | 5.88 | PRO-AYS-055 V0 2021-07- 19 Protein determination according to ISO 1871 |
| | TOTAL CARBOHY-DRATES | g/100 g | N/A | 84.05 | Calculation |
| | TOTAL CALORIES | kcal/100 g | N/A | 365.21 | Calculation |
| COF-FEE PULP RAISINS | MOISTURE AND VOLATILE MATTER | g/100 g | N/A | 26.13 | PRO-AYS-057 V0 2021-07- 19 Determination of total solids and drying losses at 103 °C y 130 °C |
| | ASH | g/100 g | N/A | 1.38 | PRO-AYS-067 V0 2021-07- 19 Ash Determination 550 °C |
| | TOTAL GRASES | g/100 g | N/A | 0.32 | PRO-AYS-113 V0 2021-07- 19 Determination of fat by ethereal extract |
| | TOTAL PROTEIN | g/100 g | N/A | 2.12 | PRO-AYS-055 V0 2021-07- 19 Protein determination according to ISO 1871 |
| | TOTAL CARBOHY-DRATES | g/100 g | N/A | 70.05 | Calculation |
| | TOTAL CALORIES | kcal/100 g | N/A | 291.56 | Calculation |

## 4. Conclusions

The production of coffee in the Department is approximately 130,625,000 kg; considering that the pulp is equivalent to 40% of the bean, the amount of pulp that can be used is 52,250,000 kg. The use of this waste contributes to sustainability, giving added value to the coffee industry, helping the economy, generating employment, and reducing the impact generated by poorly managed waste, thus improving the population's quality of life.

It was observed that the variables that most influence the fulfillment of technical specifications and the organoleptic characteristics of the final product are temperature, time, and weight; for this reason, it is indispensable to carry out a quality plan that guarantees the control and monitoring of these variables.

When interpreting each of the graphs generated by the Minitab Statistical software for the experimental mix design, it was found that the adequate formulation is constituted by 50% of cereal, 30% honeydew molasses, and 20% coffee pulp since this is the one that guarantees compliance with the specifications.

The microbiological and proximate analysis results conclude that the products are safe and suitable for human consumption. Because these did not show growth of microorganisms and addition, they provide high energy content of 365.21 and 291.56 kcal/100 g for the energy bar and coffee pulp raisins, respectively.

**Author Contributions:** Conceptualisation, A.P.C., Z.D.E. and N.P.R.; methodology, A.P.C., Z.D.E.; software, A.P.C. and N.P.R.; validation, N.P.R. and Z.D.E.; formal analysis, A.P.C., Z.D.E. and N.P.R.; writing, revising and editing, A.P.C., N.P.R. and Z.D.E. All authors have read and accepted the published version of the manuscript.

**Funding:** This research received no external funding.

**Acknowledgments:** We thank MinCiencias for the stipend to initiate the research project through the national royalty system for the year 2022.

**Conflicts of Interest:** The authors declare no conflicts of interest.

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
