# Peer review of "Coffee Pulp: A Sustainable and Affordable Source for Developing Functional Foods"

_processes, doi:10.3390/pr11061693_

Round 1

Reviewer 1 Report

The manuscript shows less interpretation of the results with non-significant experimental design. The authors should perform cross-check of the reported model and include the statistical analysis. However, I proposed the following comments:

1. Introduction should be more elaborated defining the novelty of the present work.

2. Line 161 Proximate and microbiological analysis: how this analysis was performed? The authors are requested to provided details for microbiological analysis.

3.  Line 246 P value change to p value

4. How model p value > 0.05 is significant? p value should be ≤ 0.05 for a significant model. The proposed model is not valid in the study. Authors must do the cross-check of the model. 

5. Line 242-245 "On the other hand, panela syrup and coffee pulp have an inversely proportional relationship with the caloric 243 value since when these ingredients increase their participation in the formulation, the caloric value decrease-why caloric value decreases." Explain: How it affects the diabetic and obese patients?

6. Line 270-Authors are requested to provide images of microbial analysis experiments

7. The authors are requested to perform ANOVA and report multiple comparison procedure for the experimental data in Table 4, 6 and Table 8.

Author Response

Good afternoon I am enclosing a response to your valuable comments.

Reviewer 2 Report

the ms needs revision according to my comments on the pdf attached copy.

Author Response

(The authors gave the same response as above.)

Round 2

Reviewer 1 Report

The manuscript can be accepted.